# Determining the stability of genome-wide factors in BMI between ages 40 to 69 years

Nathan A. Gillespie[1,2]*, Amanda Elswick Gentry[1], Robert M. Kirkpatrick[1], Chandra A. Reynolds[3], Ravi Mathur[4], Kenneth S. Kendler[1], Hermine H. Maes[5], Bradley T. Webb[1,4‡], Roseann E. Peterson[1‡]

**1** Virginia Institute for Psychiatric and Behavior Genetics, Department of Psychiatry, Virginia Commonwealth University, Richmond, Virginia, United States of America, **2** QIMR Berghofer Medical Research Institute, Herston, Australia, **3** Department of Psychology, University of California, Riverside, California, United States of America, **4** GenOmics, Bioinformatics, and Translational Research Center, Biostatistics and Epidemiology Division, RTI International, Research Triangle Park, North Carolina, United States of America, **5** Virginia Institute for Psychiatric and Behavior Genetics, Departments of Human and Molecular Genetics, Psychiatry, & Massey Cancer Center, Virginia Commonwealth University, Richmond, Virginia, United States of America

‡ These authors shared senior authorship on this work.
* nathan.gillespie@vcuhealth.org

**Data Availability Statement:** All BMI and GWAS data used here are publicly available from the UK Biobank (https://www.ukbiobank.ac.uk). All genetic variance-covariance (S) and asymptotic sampling

## Abstract

Genome-wide association studies (GWAS) have successfully identified common variants associated with BMI. However, the stability of aggregate genetic variation influencing BMI from midlife and beyond is unknown. By analysing 165,717 men and 193,073 women from the UKBiobank, we performed BMI GWAS on six independent five-year age intervals between 40 and 72 years. We then applied genomic structural equation modeling to test competing hypotheses regarding the stability of genetic effects for BMI. LDSR genetic correlations between BMI assessed between ages 40 to 73 were all very high and ranged 0.89 to 1.00. Genomic structural equation modeling revealed that molecular genetic variance in BMI at each age interval could not be explained by the accumulation of any age-specific genetic influences or autoregressive processes. Instead, a common set of stable genetic influences appears to underpin genome-wide variation in BMI from middle to early old age in men and women alike.

## Author summary

We used a new method called genomic structural equation modeling to analyse data from 165,717 men and 193,073 women from the UKBiobank. Our results revealed that genetic influences on BMI were very stable between ages 40 and 73. In other words, there did not appear to be any age-dependent genetic influences on BMI during this period. The same results were seen in men and women.

## Introduction

The recent decade has witnessed significant advances in the detection of multiple loci underpinning variation in complex traits [1]. Among the most successful endeavors has been

covariance 'weight' (V) matrices along with our Lavaan code are available for download at https://github.com/nathangillespie/VIPBG_public_2021_08_gSEM_BMI/.

**Funding:** BTW was supported by NIH-NIAAA (https://www.niaaa.nih.gov) (Grant number P50AA022537). REP was supported by NIH-NIAAA (https://www.niaaa.nih.gov) (Grant number P50AA022537). RMK was supported by NIH-NIAAA (https://www.niaaa.nih.gov) (Grant number P50AA022537). REP was supported by NIH-NIMH (https://www.nimh.nih.gov) (Grant number K01MH113848). REP was supported by The Brain & Behavior Research Foundation NARSAD (https://www.bbrfoundation.org) (Grant number 28632). HHM was supported by NIH-NIAAA (https://www.niaaa.nih.gov) (Grant number R01AA015416). AEG was supported by NIH-NIMH (https://www.nimh.nih.gov) (Grant number T32MH020030). CAR was supported by NIH-NIA (https://www.nia.nih.gov) (Grant numbers AG046938 & AG060470). The funders had no role in study design, data collection and analysis, decision to publish, or preparation of the manuscript.

**Competing interests:** The authors have declared that no competing interests exist.

genome-wide association scan (GWAS) analyses of adult BMI [2–4]. Notwithstanding the predictive validity of common BMI variants [5], GWAS BMI loci are based on large, aggregated meta-analytic samples derived from varying geographic and economic regions, derived from different birth cohorts and age distributions. Here, we examine this last caveat for it remains to be empirically determined if the genome-wide variation associated with adult BMI is age-invariant or age-specific. This has public-health consequences. For example, given the positive association between increased age and BMI with COVID19-related hospitalization and mortality [6,7], determining if genetic variants in BMI at older ages are qualitatively distinct is important.

## BMI heritability & longitudinal genetic correlations

Despite moderate stability across time [8–12], average adult BMI increases from age 20 to age 65 at which time it levels off until age 80 [12] before beginning to decline. Such changes might be attributed to variable contributions of genetic and environmental risks across the lifespan. Apart from birth cohort differences, Dahl et al. [12], found that factors such as an obesity genetic risk score, type-2 diabetes mellitus, cardiovascular disease, substance use, and educational attainment were all differentially predictive of both average BMI and changes in BMI before age 65. In contrast, many of these risks were no longer predictive after age 65. In terms of genetics, whereas the overall lifetime BMI heritability is 0.75 [13], heritability actually increases throughout infancy and adolescence [14] before decreasing during adulthood [13]. In terms of cross-temporal associations, genetic influences in BMI are correlated across time [8–11,14–16], sometimes very highly [17], which indicates continuous expression of the same genetic influences [18]. However, longitudinal genetic correlations for BMI never reach unity. Indeed, there is considerable variability in longitudinal genetic correlations [13,15,16]. This is also consistent with age-specific genetic influences, which could be obscured in GWAS meta-analytic results based on data aggregated across age or GWAS analyses with age as a covariate.

## Linkage Disequilibrium Score (LDSC) regression

Until recently, estimates of heritability ($h^2$) and genetic correlations ($r_G$), have typically relied on twins reared together or family studies within a structural equation modeling (SEM) framework. The development LDSC [19] has circumvented the need for twin studies by now making it possible to estimate $h^2$ and $r_G$ using unrelated and independent samples that have GWAS summary test statistics. Briefly, LDSC regression works by leveraging external linkage disequilibrium (LD) reference panels, which summarize correlations between genetic markers across the human genome, to produce genetic covariance matrices from GWAS summary statistics. In addition to estimating $r_G$, these matrices can then be used within a traditional SEM framework to test hypotheses about comorbidity, or the nature of change. In terms of BMI, this approach could be used to address the question of whether or not genetic risks in BMI are correlated across time.

Currently however, there is a paucity of molecular-based reports examining $h^2$ and $r_G$ between BMI assessed at different ages, and they have relied on different approaches to produce mixed results. For instance, Trzaskowski et al. [20] used LDSR to report a genetic correlation ($r_g = 0.86$) between BMI assessed at age 11 and 65. Winkler et. al. [21] estimated Spearman rank genetic correlations between BMI assessed in populations above and below age 50, which revealed much smaller correlations ($r_g = 0.05$ to $0.12$). Notwithstanding the need for greater precision regarding longitudinal genetic correlations, such correlations are descriptive and provide no insight regarding competing theories underlying developmental processes in BMI.

We argue that at least two theoretical mechanisms [22] can be invoked to explain observed continuity in genetic correlations. The first is a common factor process whereby common genetic or environmental factors determine the levels and rates of change in BMI over time. In this model, variances and covariances between longitudinal measures of BMI depend on individual genetic or environmental differences in growth patterns unfolding with age i.e., random growth curves [23–27]. We are aware of three twin studies that have applied genetically informative growth models to longitudinal BMI data [28–30]. Unfortunately, random growth curves do not determine the extent to which stability or changes in BMI are governed by time-invariant versus age-specific genetic influences. To address this question, the second mechanism predicts that variances and covariances are determined by random, time-specific genetic and environmental effects, which are more or less persistent over time i.e., autoregressive effects' [31–33]. Illustrated in Fig 1, this approach predicts a causal process of inertial effects, whereby BMI genetics at one time causally affect BMI at the next. We have applied this validated approach to personality [34], anxiety and depression [35,36], substance use [37] and brain aging [38]. We are aware of two reports that have tested autoregression models with respect to BMI data [9,18]. Cornes et al. [9] found evidence of distinct, age-specific genetic influences on BMI at ages 12, 14 and 16. To our knowledge, autoregressive effects have never been tested in adult BMI, especially across a wide window comprising narrow age intervals in adulthood. Fortunately, the recent, innovative application of structural equation modelling (SEM) [39,40] to LDSC regression genetic correlations [39] based on available GWAS results can now address the aforementioned gaps.

By applying genomic structural equation modeling or "genomicSEM" to BMI GWAS summary statistics from the UK Biobank [41], our aim was to determine if genetic influences across middle to early old age were best explained by age-dependent versus age-invariant processes. We also tested if alternative, more parsimonious theoretical explanations i.e., common factor models, whereby covariance between genetic influences across time could be better captured by a single latent factor [42], provided a better fit to the data. Given that standardized estimates of BMI heritability for men and women are statistically equal [13] and that there appear to be no sex differences in terms of the observed adulthood decline in heritability [43], we hypothesized that developmental processes governing changes in heritability over time likewise ought to be comparable across sex.

## Methods

All BMI and GWAS summary statistics came from the UK Biobank, a major biomedical database. The UK Biobank is a large-scale biomedical database and research resource containing genetic, lifestyle and health information from half a million UK participants. UK Biobank's database, which includes blood samples, heart and brain scans and genetic data of the 500,000 volunteer participants, is globally accessible to approved researchers who are undertaking health-related research that's in the public interest [41].

### BMI data

Described in detail elsewhere [44], weight was collected from subjects using a Tanita BC418MA body composition analyzer. Standing and sitting height measurements were collected from subjects using a Seca 240cm height measure. Body mass index (BMI) was calculated as weight divided by height squared (kg/m2). We divided the BMI data into six age intervals: 40–45; 46–50; 51–55; 56–60; 61–65; and 66–73 years. The range was based on available data whereas the number of age tranches was selected to maximize our power to choose between competing longitudinal and multivariate models without minimizing the statistical

power of the GWAS analyses at each interval. The number of subjects with complete BMI and GWAS summary statistics are shown in S1 Table.

## Genotypic data

Genotype data were filtered according to the Neale Lab pipeline, using filtration parameters and scripts publicly available from the lab GitHub [45]. Samples were filtered to retain only unrelated subjects of British ancestry (n = 359,980.) Imputed variants [46] were filtered for INFO scores > 0.8, MAF > 0.001, and HWE p-value > 1e-10.

## GWAS analyses

This is proof-of-principle illustration of the application of structural equation modelling (SEM) to GWAS summary data to test competing longitudinal hypotheses. Since the number of UKB subjects with repeated BMI measures and GWAS summary statistics was insufficient to be divided into a minimum of three age tranches needed to test the autoregression hypothesis, we treated the GWAS summary statistics at each 5-year age interval as pseudo-longitudinal. This resulted in 6 separate and independent age tranches for GWAS. Three separate GWAS analyses were conducted for each age tranche (men, women and combined) using the BGENIE (version 1.3.) [46]. The first 20 ancestry principal components were included as covariates in all models. Sex was included as a covariate in the combined (men + women) model.

## Genomic structural equation modelling

We then applied the GenomicSEM software package [39] in R (version 4.0.3) [47] to the BMI GWAS results to estimate separate genetic variance-covariance (S) and asymptotic sampling covariance 'weight' (V) matrices for the male, female and then the combined GWAS results.

Estimation of the S and V matrices is a 3-step process. In step 1, the raw GWAS summary statistics were manipulated using the GenomicSEM *munge* option to remove all SNPs with MAF < 1%, information scores < 0.9, and SNPs in the MHC region. In step 2, we used the GenomicSEM *ldsc* option to run multivariate LD score regression [39] to estimate the S and V matrices between the GWAS summary statistics. This method has been successfully applied to detect genetic correlations between bio-medical, psychiatric and behavioural phenotypes [48–68], which are commensurate with previous biometrical genetic correlations [69–76] while revealing extensive pleiotropy across a wide variety of phenotypes. In step 3, the S and V matrices were then read into the lavaan (version 0.6–7) [77] SEM software package in R (version 4.0.3) [47] to fit and compare competing longitudinal and multivariate models. All Genomic-SEM and lavaan scripts used here are publicly available at https://github.com/ToddWebb/UKBiobank_VIPBG/tree/master/LongitudinalGenomicSEM.

The autoregression model predicts that time-specific random genetic or environmental effects are more or less persistent over time (autoregressive effects) [31]. As described by Eaves and others [31–33], autoregression models assume that the covariance structure arises because of random, age-specific genetic or environmental effects, which are, at least partially, carried forward. As illustrated in Fig 1, innovations at each assessment reflect novel, time- or age-specific genetic or environmental influences, which are uncorrelated with previous genetic influences. Genetic differences at each occasion are therefore a function of new random effects on the phenotype that arise as well as a (linear) function of individual genetic differences expressed at the preceding time. Here, we assume that cross-temporal correlations arise because the innovations have a more or less persistent effect over time and may, under some circumstances accumulate, potentially giving rise to developmental increases in genetic variance and increased correlations between adjacent measures. One consequence of the

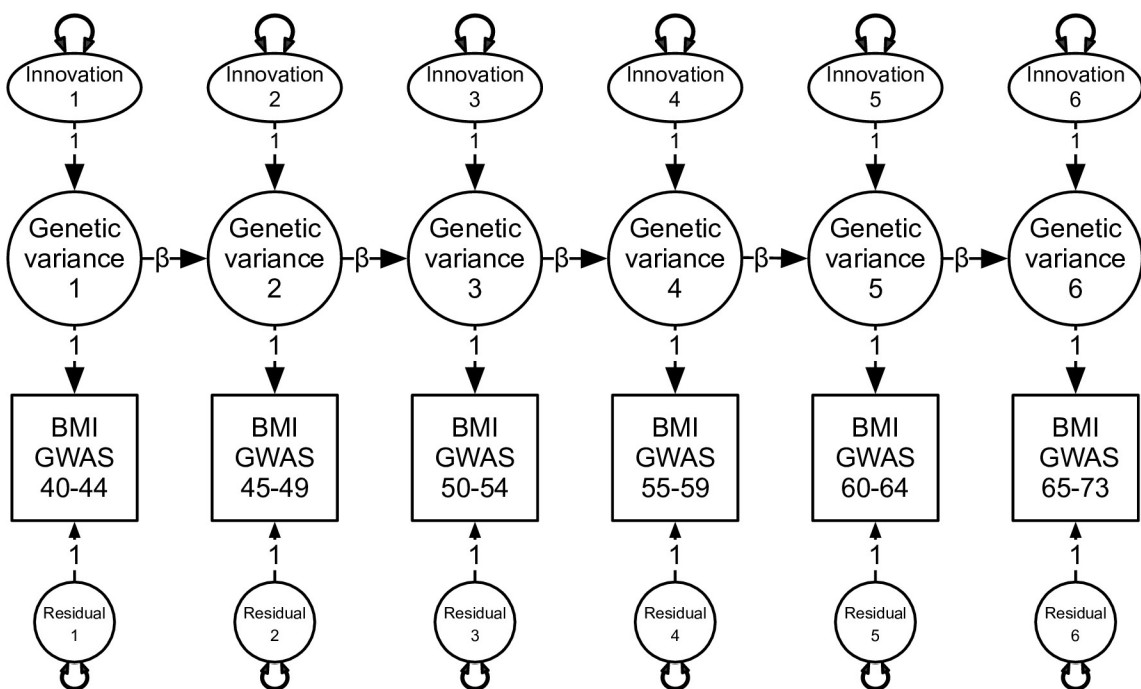

**Fig 1. Autoregression model depicting genome-wide variation in BMI at each age interval.** This development model predicts that genetic variation at each time interval can be decomposed into time-specific variation or 'innovations' & the causal contribution of genome-wide genetic variation from previous age intervals. Innovations refer to novel or age-specific genetic influences that are uncorrelated with previous genetic influences. This model also includes residual genetic variation not otherwise explained by the autoregression process. Double-headed arrows denote variation associated with innovations & residuals at each age interval. Beta (β) denotes the causal contribution of genetic variance from one age interval to the next.

autoregressive model is the tendency of cross-temporal correlations to decay as a function of increasing lag-time. Depending on the magnitude of an innovation and its relative persistence, the observed variances and cross-temporal covariances may increase towards a stable asymptotic value. We began by fitting innovations at all six time-intervals, which were then successively dropped. We also specified an autoregression model that included a single innovation at BMI 40–44 accounting for all subsequent genetic variances. Technically, this first innovation includes all genetic variance accumulated up to this first age interval. Finally, we fitted a factor analysis comprising a single factor.

### Model fit indices & comparisons

In GenomicSEM analyses there is no one sample size to speak of. This is because GWAS studies from which the summary statistics are derived can vary in size and subject overlap. Thus, potentially, a different (effective) sample size may apply to each element of S. We were therefore limited to fit indices that do not explicitly depend upon sample size: the pseudo Akaike Information Criterion (pseudoAIC); Comparative Fit Index (CFI); Tucker Lewis Index (TFI); and the Standardized Root Mean Square Residual (SRMR) to judge the best-fitting model. Both the CFI and TFI are incremental fit indices that penalize models with increasing complexity. The SRMR is an absolute measure of fit based on the difference between the observed and predicted correlations under each model, such that a value of zero indicates a perfect fit. The pseudoAIC is a comparative fit index, whereby the model with the lowest AIC values is interpreted as the best-fitting.

**Table 1. Sample sizes, estimates of SNP-based heritability (including standard errors along diagonal) & linkage disequilibrium score regression genetic correlations between the six age intervals based on the combined male & female GWAS summary statistics.**

|  | Sample size | 1. | 2. | 3. | 4. | 5. | 6. |
|---|---|---|---|---|---|---|---|
| 1. BMI GWAS 40–44 yrs | 34,001 | 0.23 (0.02) |  |  |  |  |  |
| 2. BMI GWAS 45–49 yrs | 45,294 | 1.00 | 0.26 (0.02) |  |  |  |  |
| 3. BMI GWAS 50–54 yrs | 53,602 | 0.99 | 1.00 | 0.26 (0.02) |  |  |  |
| 4. BMI GWAS 55–59 yrs | 64,891 | 0.93 | 0.93 | 0.95 | 0.29 (0.01) |  |  |
| 5. BMI GWAS 60–64 yrs | 89,824 | 0.95 | 0.94 | 0.93 | 0.90 | 0.24 (0.01) |  |
| 6. BMI GWAS 65–73 yrs | 71,178 | 0.97 | 0.96 | 0.95 | 0.93 | 1.00 | 0.22 (0.01) |

## Results

### Combined male & female analyses

The LDSR-based genome-wide genetic correlations between the six GWAS summary statistics, including GWAS sample sizes and the SNP-based heritability for each age interval, are shown in Table 1. The correlations do not decline with increasing time intervals, which would be indicative of a simplex structure best explained by autoregression models. For example, the LDSR genetic correlation ($r_g$) between BMI at ages 40–45 and 66–73 years was higher than the $r_g$ between BMI at ages 40–45 and 56–60 years ($r_g = 0.97$ vs 0.93). Overall, the genetic correlations were very high and ranged from 0.93 to 1.00.

Formal model fitting comparisons are shown in Table 2. We began with a fully saturated autoregression model comprising unique genetic influences or innovations at each age interval (Fig 1). This provided a reasonable fit to the data as judged by the non-significant chi-square, very high CFI and TLI values and very low SRMR. Autoregression sub-models in which the genetic innovations at ages 66 to 73, 61 to 65, and 51 to 55 years were each successively removed provided only marginal improvements in terms of their pseudoAIC values. In contrast, the factor analysis with a single factor provided the overall best fit in terms of the smallest chi-square, lowest pseudoAIC and lowest SRMR. In this model (see Fig 2), genetic variance at each five-year age interval was best explained by a single factor with a genome-wide SNP heritability of 24%.

### Sex specific analyses

An identical pattern emerged when the model fitting was repeated by sex. Male and female sample sizes at each age interval are shown in S1 Table. Table 3 shows the LDSR genetic correlations for men and women. Varying only slightly, the separate male and female correlations

**Table 2. Multivariate modeling fitting comparisons based on the combined male & female GWAS summary statistics.**

| Models | Chi-square(df) | p | pseudoAIC | CFI | TLI | SRMR |
|---|---|---|---|---|---|---|
| Full auto-regression (AutoReg) | 21.113(13) | 0.071 | 55.113 | 0.999 | 0.999 | 0.039 |
| AutoReg: genetic innovation at 65–73 yrs dropped | 22.419(14) | 0.070 | 54.419 | 1.000 | 0.999 | 0.039 |
| AutoReg: genetic innovation at 60–64 yrs dropped | 20.872(14) | 0.105 | 52.872 | 1.000 | 0.999 | 0.039 |
| AutoReg: genetic innovation at 55–59 yrs dropped | 25.403(14) | 0.031 | 57.403 | 0.999 | 0.999 | 0.041 |
| AutoReg: genetic innovation at 50–54 yrs dropped | 21.768(14) | 0.084 | 53.768 | 0.999 | 0.999 | 0.040 |
| AutoReg: genetic innovation at 45–49 yrs dropped | 34.073(14) | 0.002 | 66.073 | 0.998 | 0.998 | 0.051 |
| AutoReg: genetic innovations at 45–73 yrs dropped | 46.133(14) | 0.000 | 70.133 | 0.998 | 0.998 | 0.056 |
| Factor analysis—1 factor | 13.005(9) | 0.162 | 37.005 | 1.000 | 1.000 | 0.016 |

Note: AIC = Akaike Information Criterion, CFI = Comparative Fit Index, TLI = Tucker Lewis Index, SRMR = (Standardized) Root Mean Square Residual. Innovations refer to novel or age-specific genetic influences that are uncorrelated with previous genetic influences.

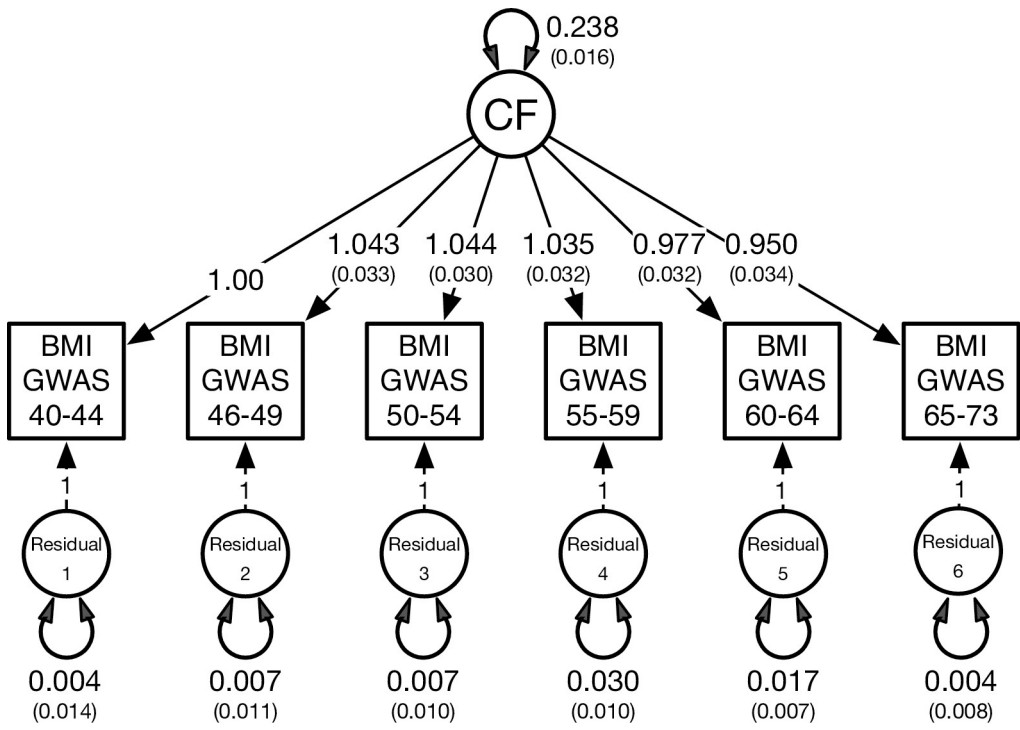

**Fig 2. Best fitting factor analytic model with a single common factor (CF) based on the combined male and female data.** The CF explains covariation between the six GWAS summary statistics each based on five-year intervals between ages 40–73 years. To identify this model, the first factor loading from the CF to BMI GWAS at 40–45 years was constrained to one. The double-headed arrow on the CF denotes the standardized variance, or SNP-based heritability, for BMI. Double-headed arrows on the residuals denote genetic variation at each age interval not otherwise explained by the CF.

were again high and ranged from $r_g = 0.88$ to $r_g = 1.00$. S1 Table also shows the SNP-based heritability estimates by sex, which were very similar at each age interval.

As shown in Table 4, the genetic innovations at ages 46 to 66+ years for men and women could each be dropped from the full autoregression model as judged by the non-significant chi-square value. Overall, for both sexes, the factor analysis with a single common factor again provided the best fit to the data in terms of the lowest chi-square, pseudoAIC and SRMR values (see Fig 3). This suggests that there is no evidence of age-specific genome-wide variation in BMI for either men or women.

## Discussion

This is the first study to test a developmental theory regarding BMI heritability using molecular data and structural equation modeling. Between ages 40 and 73, changes in BMI heritability

**Table 3. Linkage disequilibrium score regression genetic correlations based on the male (below diagonal) & female (above diagonal italics) GWAS summary statistics at six age intervals.**

|  | 1. | 2. | 3. | 4. | 5. | 6. |
|---|---|---|---|---|---|---|
| 1. BMI GWAS 40–44 yrs | 1 | *0.99* | *0.99* | *0.91* | *0.95* | *0.92* |
| 2. BMI GWAS 45–49 yrs | 0.98 | 1 | *1.00* | *0.96* | *0.93* | *0.93* |
| 3. BMI GWAS 50–54 yrs | 1.00 | 0.99 | 1 | *0.95* | *0.91* | *0.90* |
| 4. BMI GWAS 55–59 yrs | 0.93 | 0.89 | 0.93 | 1 | *0.88* | *0.93* |
| 5. BMI GWAS 60–64 yrs | 0.97 | 0.90 | 0.95 | 0.88 | 1 | *0.98* |
| 6. BMI GWAS 65–73 yrs | 0.97 | 0.90 | 0.96 | 0.93 | 0.99 | 1 |

**Table 4. Multivariate modeling fitting comparisons based on the combined MALE GWAS summary statistics at six age intervals.**

| Women | ChiSquare$_{df}$ | p | pseudoAIC | CFI | TLI | SRMR |
|---|---|---|---|---|---|---|
| Full auto-regression (AutoReg) | 15.019$_{(13)}$ | 0.306 | 49.019 | 1.000 | 1.000 | 0.043 |
| AutoReg: genetic innovation at 65–73 yrs dropped | 14.866$_{(14)}$ | 0.387 | 46.866 | 1.000 | 1.000 | 0.043 |
| AutoReg: genetic innovation at 60–64 yrs dropped | 14.883$_{(14)}$ | 0.386 | 46.883 | 1.000 | 1.000 | 0.043 |
| AutoReg: genetic innovation at 55–59 yrs dropped | 16.813$_{(14)}$ | 0.266 | 48.813 | 0.999 | 0.999 | 0.046 |
| AutoReg: genetic innovation at 50–54 yrs dropped | 14.213$_{(14)}$ | 0.434 | 46.213 | 1.000 | 1.000 | 0.043 |
| AutoReg: genetic innovation at 45–49 yrs dropped | 21.482$_{(14)}$ | 0.090 | 53.482 | 0.998 | 0.998 | 0.057 |
| AutoReg: genetic innovations at 45–73 yrs dropped | 25.617$_{(14)}$ | 0.109 | 49.617 | 0.998 | 0.999 | 0.059 |
| Factor analysis—1 factor | 8.832$_{(9)}$ | 0.453 | 32.832 | 1.000 | 1.000 | 0.023 |
| Men | | | | | | |
| Full auto-regression (AutoReg) | 11.858$_{(13)}$ | 0.539 | 45.858 | 1.000 | 1.000 | 0.054 |
| AutoReg: genetic innovation at 65–73 yrs dropped | 12.814$_{(14)}$ | 0.541 | 44.814 | 1.000 | 1.000 | 0.054 |
| AutoReg: genetic innovation at 60–64 yrs dropped | 12.085$_{(14)}$ | 0.599 | 44.085 | 1.000 | 1.000 | 0.053 |
| AutoReg: genetic innovation at 55–59 yrs dropped | 11.889$_{(14)}$ | 0.615 | 43.889 | 1.000 | 1.000 | 0.053 |
| AutoReg: genetic innovation at 50–54 yrs dropped | 21.826$_{(14)}$ | 0.082 | 53.826 | 0.998 | 0.998 | 0.064 |
| AutoReg: genetic innovation at 45–49 yrs dropped | 12.718$_{(14)}$ | 0.549 | 44.718 | 1.000 | 1.000 | 0.059 |
| AutoReg: genetic innovations at 45–73 yrs dropped | 1628.983$_{(14)}$ | 0.000 | 1652.983 | 0.168 | 0.001 | 0.803 |
| Factor analysis—1 factor | 4.398$_{(9)}$ | 0.883 | 28.398 | 1.001 | 1.000 | 0.018 |

Note: AIC = Akaike Information Criterion, CFI = Comparative Fit Index, TLI = Tucker Lewis Index, SRMR = (Standardized) Root Mean Square Residual

could not be explained by detectable age-specific genome-wide variation or an accumulation of genetic variants over time. Instead, individual differences in the molecular genetics of BMI across this time span were best explained by a single or common set of stable, genetic influences that are observable in early midlife. This pattern was observed in men and women.

Our results are consistent with Silventoinen et al.'s meta-analysis of twin data that revealed only minor differences in BMI heritability estimates across cultural-geographic regions and measurement time [78,79]. Dahl et al.'s [12] analysis of Swedish twin data revealed that for men and women, BMI increases across midlife, before leveling off at 65 years and declining at approximately age 80. The extent to which Dahl et al.'s observed inflexion at age 65 years is indicative of age-specific, distinct genetic influences or variance components was inconsistent with our results. Instead, we found that the genetic correlations between BMI at ages 61–65 years and the remaining four age tranches were all equally high. Thus, the molecular genetic variance at age 65 does not appear to be linked to age-specific or distinct genetic processes occurring around this time.

An outstanding question is whether or not our results generalize to earlier stages in life. Here, a number of reports, relying on different methods, suggest that genetic risks spanning childhood, adolescence and early adulthood likely comprise a combination of age-specific and age-invariant influences. For example, several studies have shown that the PRS for childhood BMI predicts adult BMI, metabolic outcomes and other complex traits [80–84]. Other studies relying on twin data have reported genetic correlations between BMI assessed at shorter age intervals spanning infancy, adolescence and teenage years that are much higher compared to genetic correlations based on wider age intervals [8,14,85,86]. This pattern is consistent with autoregressive features. Cornes et al.'s [9] application of autoregressive modelling found evidence of largely age-invariant genetic influences on BMI at 12, 14 and 16. The authors also reported smaller but significant age-specific genetic influences on BMI, depending on sex, at 14 and 16 years that were uncorrelated with BMI at age 12 [9]. The pattern of age-invariant

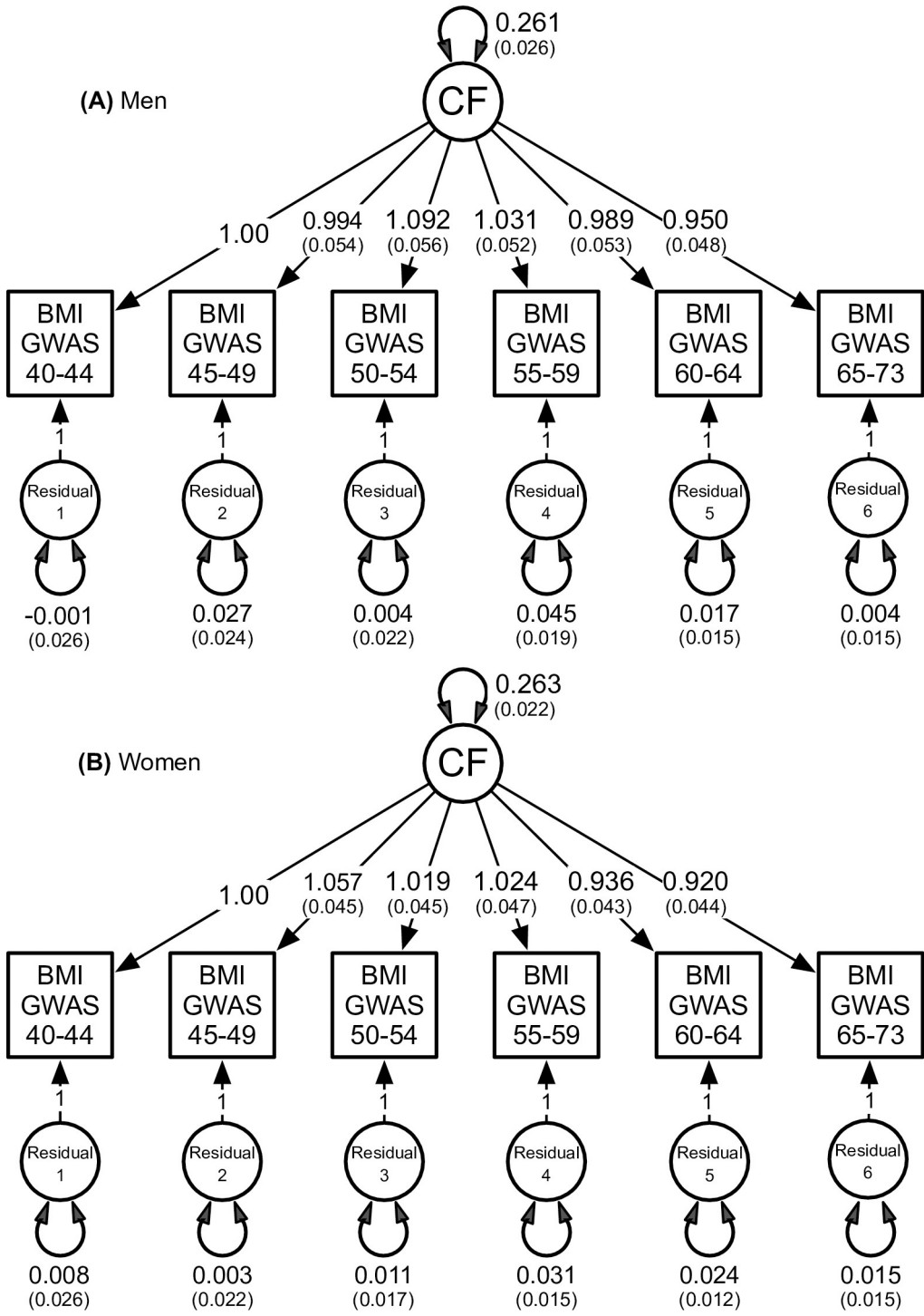

**Fig 3. Best fitting factor analytic model with a single common factor (CF) for men** (A) **& women** (B). To identify this model, the first factor loading from the CD to BMI GWAS at 40–45 years was constrained to one. The double-headed arrow on the CF denotes the standardized variance, or SNP-based heritability, for BMI. Double-headed arrows on the residuals denote genetic variation at each age interval not otherwise explained by the CF.

influences is consistent with the study by Couto Alves et al. [87], which examined BMI spanning ages 2 to 18 years and reported a robust overlap between the genetics of child and adult BMI. The same study also identified a completely distinct genetic architecture in infancy [87]. The reports by Warrington et al. [88] and Felix et al. [89] have shown how numerous replicated adult BMI loci also reach genome-wide significance in childhood GWAS studies of BMI. In terms of other LDSR studies, Trzaskowski et al. [20] reported a genetic correlation between BMI at 11 and 65 years of 0.86. The same study also found that the adult PRS for BMI explained at most 10% of the phenotypic variance in childhood BMI. Combined, the findings based on twin and molecular studies suggest that variability in heritability estimates spanning childhood, adolescence and early adulthood is likely explained by combination of mostly age-invariant plus age-specific genetic influences, which could potentially be better captured by autoregressive modeling.

## Limitations

Our findings should be interpreted in the context of four limitations.

First, the BMI data used here were not repeated measures, but pseudo longitudinal. This approach assumes no year of birth or cohort-related genetic heterogeneity. Until now, GenomicSEM reports have typically leveraged LDSR-derived genetic covariances in the context of cross-sectional hypotheses. Our pseudo longitudinal modeling is not unlike standard cross-sectional GenomicSEM analyses. Both approaches depend upon the GWAS summary statistics being derived from a homogenous ancestral group. There is also no requirement for summary statistics to be based on the same subjects. It remains important to reduce the likelihood that our age-specific GWAS results comprised subjects from heterogenous populations. This is important because cohorts can have different BMI heritability, different environmental influences on BMI, or differences in the genetic control of sensitivity to the environment, which can bias the covariance estimates. Danish and Swedish twin studies have illustrated differential heritability by showing how increases in mean BMI in successively younger cohorts has been accompanied by increasing genetic variance [90,91]. Therefore, to determine if cohort effects existed, we inspected the LDSR genetic correlations between the youngest and oldest age tranches i.e., two maximally age-discrepant samples of unrelated individuals. Here, the $r_g$ was 0.97 (see Table 1), which suggests that the likelihood of any cohort-related genetic heterogeneity was minimal. Note, we also performed GWAS on a subset of men (N = 8,337) and women (N = 7,681) with repeated BMI measures at any time from 48 to 61 years and then from 62 to 72 years. Here, the test-rest LDSR correlations were $r_g$ = 0.99 (p = 1.04e-03) and $r_g$ = 0.95 (p = 1.68e-04) for men and women respectively. Thus, genetic correlations were very high between *and* within subjects.

In another attempt to reduce the possibility of systematic differences between putative sub-populations in terms of allele frequencies, we re-ran the GWAS analyses using 40 PCs as covariates. As shown in S1 Table, there were only very minor differences in genomic inflation and SNP heritability between the 20 versus 40 PC results. Therefore, not only did genomic inflation remain the same regardless of the number of PCs, it did not change across age intervals. These results further reduce the likelihood of birth or cohort-related genetic heterogeneity.

Second, the UKB recruitment process did not represent a random sample of the UK population [92]. Subjects were predominately European, more likely to be older, female, to live in less socioeconomically deprived areas than nonparticipants, and when compared with the general population, were also less likely to be obese, to smoke, and to drink alcohol daily while reporting fewer self-reported health conditions [93,94]. Although Silventoinen et al.'s meta-analysis of twin data reported only minor differences in BMI heritability across divergent

cultural-geographic regions [78,79], the extent to which the molecular-based genetic covariance structure observed here generalizes to non-European populations remains to be determined.

Third, while our results illustrate the flexibility of SEM in terms of its application to GWAS summary statistics to test a theory of longitudinal change, our modeling was not exhaustive. For instance, we did not test the hypothesis that changes in heritability could be better explained by latent growth or latent growth mixture models [95,96]. We note that the current method is limited to the analysis of summary variance-covariance matrices derived from the analysis of common variants. GenomicSEM does not model observed phenotypic information. Consequently, there was no mean information to model latent growth or mixture distributions. We also did not test hypotheses regarding sex differences other than to report results by sex. Dubois' meta-analysis of 23 twin birth-cohorts found evidence of sex-limitation in terms of greater genetic variance in boys in early infancy through to 19 years [97]. In contrast, Elks et al.'s meta-regression of 88 twin-bases estimates of BMI heritability found no evidence of sex effects [13]. It remains to be determined if the observed minor differences in the genetic covariances and the ultimate, best fitting single-factor structure are empirically equivalent across sex.

Finally, our genomic modelling was based on aggregated GWAS summary data and so was entirely independent of environmental risks, which are known to be significant in the etiology of psychiatric and behavioral traits [98]. Consequently, our current approach precludes modeling the contribution of environmental influences with increasing age [99] or making allowances for any genetic control of sensitivity to the environment i.e. G x E interaction [78]. In this regard, methods that can simultaneously model the joint effect of genes and environment are likely to prove more informative. For instance, innovative approaches capable of applying genomic-relatedness based restricted maximum-likelihood [100] to structural equation modeling in the OpenMx [40] software package have the potential to analyze individual GWAS and phenotypic data and hold promise.

## Conclusion

Structural equation model of GWAS summary statistics between the ages of 40 and 73 revealed that molecular genetic variance in BMI at successive 5-year age intervals could not be explained by the accumulation of age-specific genetic influences or autoregressive processes. Instead, a common set of stable genetic influences appears to underpin all genome-wide variation in BMI from middle to early old age in men and women.

## Supporting information

**S1 Table. Number of men & women with complete BMI & GWAS summary statistics at each age interval, as well as genomic inflation (λ) & SNP-based heritability ($h^2$) for the GWAS analyses comprising 20 versus 40 principal components (PCs).**
(DOCX)

## Author Contributions

**Conceptualization:** Nathan A. Gillespie, Chandra A. Reynolds, Kenneth S. Kendler.

**Data curation:** Amanda Elswick Gentry, Bradley T. Webb.

**Formal analysis:** Nathan A. Gillespie, Robert M. Kirkpatrick.

**Methodology:** Nathan A. Gillespie, Robert M. Kirkpatrick.

**Writing – original draft:** Nathan A. Gillespie, Amanda Elswick Gentry, Robert M. Kirkpatrick, Chandra A. Reynolds, Ravi Mathur, Kenneth S. Kendler, Hermine H. Maes, Bradley T. Webb, Roseann E. Peterson.

**Writing – review & editing:** Nathan A. Gillespie, Amanda Elswick Gentry, Robert M. Kirkpatrick, Chandra A. Reynolds, Ravi Mathur, Kenneth S. Kendler, Hermine H. Maes, Bradley T. Webb, Roseann E. Peterson.

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
