## [Decision Letter · Decision Letter 0]

9 Nov 2021

Dear Dr Gillespie,

Thank you very much for submitting your Research Article entitled 'Determining the stability of genome-wide factors in BMI between ages 40 to 69 years.' to PLOS Genetics.

The manuscript was fully evaluated at the editorial level and by independent peer reviewers. The reviewers appreciated the attention to an important problem, but raised some substantial concerns about the current manuscript. Based on the reviews, we will not be able to accept this version of the manuscript, but we would be willing to review a much-revised version. We cannot, of course, promise publication at that time.

Because many readers may not be familiar with gSEMs, we would like to see a rewritten method  with more elucidation. Additional analyses as suggested by reviewer 2 should be included.

If you decide to revise the manuscript for further consideration at PLOS Genetics, please aim to resubmit within the next 60 days, unless it will take extra time to address the concerns of the reviewers, in which case we would appreciate an expected resubmission date by email to plosgenetics@plos.org.

[LINK]

We are sorry that we cannot be more positive about your manuscript at this stage. Please do not hesitate to contact us if you have any concerns or questions.

Yours sincerely,

Xiaofeng Zhu

Associate Editor

PLOS Genetics

Scott Williams

Section Editor: Natural Variation

PLOS Genetics

Reviewer's Responses to Questions

**Comments to the Authors:**

Reviewer #1: Gillespie et al perform a a valuable and insightful structural analysis of age stratified GWAS of BMI, effectively fitting a pseudo longitudinal model to cross sectional data. It is exciting to see GenomicSEM being used in a creative and informative manner.

The results reflect very little novel genetic effects on BMI in older adults in the UKB, this is not unexpected as its generally the age period in which the rank order of BMI in a popualtion across time is more stable than during development or in old age.

Id like too begin the review by mentioning some essential papers that do related things, without undermining the novelty of the current work, but that should be discussed or tied to the current effort.

Here is work using PRS and/or multivariable MR and "BMI" age age 10 and in adulthood (I am not a author on any of these, just all very relevant lit):

Richardson, T. G., Sanderson, E., Elsworth, B., Tilling, K., & Smith, G. D. (2019). Can the impact of childhood adiposity on disease risk be reversed? A Mendelian randomization study. medRxiv, 19008011.

Brandkvist, M., Bjørngaard, J. H., Ødegård, R. A., Åsvold, B. O., Smith, G. D., Brumpton, B., ... & Vie, G. Å. (2020). Separating the genetics of childhood and adult obesity: a validation study of genetic scores for body mass index in adolescence and adulthood in the HUNT Study. Human molecular genetics, 29(24), 3966-3973.

Richardson, T. G., Crouch, D. J., Power, G. M., Berstein, F. M., Hazelwood, E., Fang, S., ... & Smith, G. D. (2021). Disentangling the direct and indirect effects of childhood adiposity on type 1 diabetes and immune-associated diseases: a multivariable Mendelian randomization study. medRxiv.

There is more work on repeatedly measured BMI in MoBa, which also is very relevant.

Hone, L., Jacobs, B. M., Marshall, C. R., Giovannoni, G. R., Noyce, A., & Dobson, R. (2021). Age-specific effects of childhood BMI on multiple sclerosis risk: a Mendelian Randomisation study. medRxiv.

Helgeland, Ø., Vaudel, M., Juliusson, P. B., Holmen, O. L., Juodakis, J., Bacelis, J., ... & Njølstad, P. R. (2019). Genome-wide association study reveals dynamic role of genetic variation in infant and early childhood growth. Nature communications, 10(1), 1-10.

There are more, I feel it exceeds my responsibility as a reviewer to look those up.

With those pre-requisite out of the way, lets get to the paper itself, mayor concerns:

You conclude that: "differences in BMI across age could not be explained by the accumulation of age-specific genetic influences or autoregressive processes" This is true but it would be good to reflect on what, if anything, the genetic model selected based on parsimony would mean for the phenotypic model (if anything). I for one suspect most people would conceptualise causal auto-regressive effects of BMI at the phenotypic level, would your defining challenge such a model under what conditions would it/wouldn't it?

There are actual repeated measured within UKB, 20k people went for a second clinic visit and 46k people went in for an imaging visit. your model makes predictions about the rg we expect to see there it would be good to test those. (I admit I expect very little based on your high rg, which makes for an interesting test, what if these data do not conform?)

There is a further possible source of corroboration you could pursuit, your model implies a certain rg between BMI and change in weight, UKB includes a Q on recent weight change: https://biobank.ndph.ox.ac.uk/showcase/field.cgi?id=2306 and for those pprepeatedly measured you can compute actual weight change.

The final GWAS I listed previously among the literature suggestions has sumstats available, we have been running simple models with those, without a clear goal in mind other than to develop some instructional examples, but you could consider integrating them into this paper into a single model (then age 6 weeks to age 70...), might be beyond the scope of this manuscript. This is actually specifically a point where GenomicSEM would complement full data SEM/gSEM, genomicSEM can go beyond a single cohort, raw data SEM can allow for additional types of models.

The Genome wide model need not hold for individual GWAS hits, can you validate well know BMI loci (FTO etc) effect the age stratified outcomes in a manner consistent with the global model (You could use Q-statistic or inspect the effect sizes for various top hits)

The discussion could use a clearer discussion of the limitation of pseudo longitudonal modeling in general, when do we expect it to fail etc.

Reviewer #2: The authors applied SEMs to evaluate the stability of genome wide determinants of BMI in adult populations. The scientific question is good and clearly explained. While this is a worthwhile investigation, there are a few limitations that may make this manuscript less relevant, as it is, for the wide audience of readers interested in genetics in general.

1. The age range is limited -- only adults. It is not clear that the results is surprising given that. Previous results (Winkler et al) found lower genetic correlation between age groups of less and more than 50. It would be useful to look at younger individuals.

2. The modeling approach using gSEMs. Many researchers are not familiar with that and the specialized nomenclature used make it difficult to assess the methodology. It would be very useful-- and required -- to spell out the statistical model to explain things, e.g., what "innovations" are.

3. Also, it is not clear how the GWAS were used. Only for computing genetic correlations and heritability? (maybe it was clear, I just want to make sure). Then the Lavann software used these to construct and over all covariance matrix?

4. I was expecting to see some results estimating genetic associations of specific variants known to be highly associated with BMI, meaning, to look at their effects over time.

5. I also expected to see PRS associations evaluated over time. The authors can also construct PRS based on each of the GWAS and study associations with different age groups.

**Have all data underlying the figures and results presented in the manuscript been provided?**

Reviewer #1: **No: **Please also upload GWAS sumstats to a public repository.

Reviewer #2: None

PLOS authors have the option to publish the peer review history of their article (what does this mean?). If published, this will include your full peer review and any attached files.

Reviewer #1: No

Reviewer #2: **Yes: **Tamar Sofer

---

## [Decision Letter · Decision Letter 1]

21 Jun 2022

Dear Dr Gillespie,

We are pleased to inform you that your manuscript entitled "Determining the stability of genome-wide factors in BMI between ages 40 to 69 years." has been editorially accepted for publication in PLOS Genetics. Congratulations!

Yours sincerely,

Xiaofeng Zhu

Section Editor: Methods

PLOS Genetics

Scott Williams

Section Editor: Human Variation

PLOS Genetics

Comments from the reviewers (if applicable):

Reviewer's Responses to Questions

**Comments to the Authors:**

Reviewer #1: My comments were addressed, pseudo longitudinal modeling is an exciting potential application of genomicSEM.

Reviewer #2: Thank you for responding to the earlier comments, I have no further comments. Nice work!

**Have all data underlying the figures and results presented in the manuscript been provided?**

Reviewer #1: **No: **The underlying data is restricted access for good reasons (genotypes privacy etc etc). Qualified researchers in academia and industry can obtain access to the underlying raw data.

Reviewer #2: None

PLOS authors have the option to publish the peer review history of their article (what does this mean?). If published, this will include your full peer review and any attached files.

Reviewer #1: No

Reviewer #2: No

**Data Deposition**

http://datadryad.org/submit?journalID=pgenetics&manu=PGENETICS-D-21-01158R1

**Press Queries**

---

## [Editor Report · Acceptance letter]

26 Jul 2022

PGENETICS-D-21-01158R1 

Determining the stability of genome-wide factors in BMI between ages 40 to 69 years. 

Dear Dr Gillespie, 

We are pleased to inform you that your manuscript entitled "Determining the stability of genome-wide factors in BMI between ages 40 to 69 years." has been formally accepted for publication in PLOS Genetics! Your manuscript is now with our production department and you will be notified of the publication date in due course.

With kind regards,

Zsofi Zombor

PLOS Genetics

On behalf of:
